# Upcycling of Spent NiMH Battery Material—Reconditioned Battery Alloys Show Faster Activation and Reaction Kinetics than Pristine Alloys

**DOI:** 10.3390/molecules25102338

**Published:** 2020-05-17

**Authors:** Yang Shen, Erik Svensson Grape, Dag Noréus, Erika Widenkvist, Stina Starborg

**Affiliations:** 1Department of Materials and Environmental Chemistry, Stockholm University, SE-106 91 Stockholm, Sweden; yang.shen@nilar.com (Y.S.); erik.grape@mmk.su.se (E.S.G.); 2Nilar AB, Box 8020, SE-800 08 Gävle, Sweden; erika.widenkvistzetterstrom@nilar.com (E.W.); stina.starborg@nilar.com (S.S.)

**Keywords:** metal hydride, NiMH batteries, regeneration, reconditioning, sonication, ball-milling, acid washing, alkaline washing

## Abstract

During formation and cycling of nickel–metal hydride (NiMH cells), surface corrosion on the metal hydride particles forms a porous outer layer of needle-shaped rare-earth hydroxide crystals. Under this layer, a denser but thinner oxidized layer protects the inner metallic part of the MH electrode powder particles. Nano-sized nickel-containing clusters that are assumed to promote the charge and discharge reaction kinetics are also formed here. In this study, mechanical treatments are tested to recycle hydrogen storage alloys from spent NiMH batteries. This removes the outer corroded surface of the alloy particles, while maintaining the catalytic properties of the surface. Scanning electron microscopy images and powder X-ray diffraction measurements show that the corrosion layer can be partly removed by ball milling or sonication, combined with a simple washing procedure. The reconditioned alloy powders exhibit improved high rate properties and activate more quickly than the pristine alloy. This indicates that the protective interphase layer created on the alloy particle during their earlier cycling is rather stable. The larger active surface that is created by the mechanical impact on the surface by the treatments also improves the kinetic properties. Similarly, the mechanical strain during cycling cracks the alloy particles into finer fragments. However, some of these particles form agglomerates, reducing the accessibility for the electrolyte and rendering them inactive. The mechanical treatment also separates the agglomerates and thus further promotes reaction kinetics in the upcycled material. Altogether, this suggests that the MH electrode material can perform better in its second life in a new battery.

## 1. Introduction

Rechargeable batteries are not as rechargeable as we often would like to believe. When a battery is charged or discharged with a non-zero current, all sorts of unfavorable gradients set in. These include temperature gradients, concentrations gradients, voltage gradients and current gradients. They form over complex interfaces in the battery cell, between electrodes and electrolyte, between conductors and electrode materials, between the charged and discharged parts of the electrode, etc. These gradients drive a variety of unwanted and complicated interconnected parasitic side reactions. The inability to fully reverse all the parasitic reactions is a prerequisite for creating a chaotic system, and this will tend to make rechargeable batteries unstable with respect to charge/discharging. Cycling rechargeable batteries have something in common with forecasting weather, in that it works well in the beginning but gets progressively more difficult the further it goes. This difficulty is also reflected in the very few numbers of commercially successful rechargeable battery chemistries developed over time. In the mid-nineteenth century, the common lead–acid chemistry was developed. The two electrode materials were lead metal and lead oxide, both forming lead sulfate with sulfuric acid electrolyte during discharge, i.e., heavy and poisonous—why not select more convenient materials as almost any redox reaction can be made into an electrochemical cell? The answer is that it is very difficult to find a reversible redox reaction that is free of troublesome parasitic side reactions. We had to wait 50 years until the beginning of the last century for the nickel–iron (NiFe) battery to appear, also known as the Edison battery. Unfortunately, the iron electrode had a side reaction, forming hydrogen gas during charge. This side reaction consumed the alkaline electrolyte, making it necessary to frequent refill the battery electrolyte. Different additives were tested to mitigate the hydrogen-producing side reaction, one of which was cadmium (Cd). In the end, all iron was substituted with Cd, and the NiCd battery was created. Today, the volume of NiFe battery production volume is limited. The NiCd battery is a well-functioning battery, yet cadmium is considered a poisonous metal and there are restrictions regarding its use. Another limitation is that cadmium is a fairly rare element. The global production of cadmium is only slightly larger than that of silver. Hence, after 50 years of development, we still had only two commercially successful rechargeable battery chemistries, both of which are based on troublesome metals. Then, we had to wait almost another century for the next successful rechargeable battery chemistries: first, nickel–metal hydride, (NiMH) and, a few years later, Li batteries. NiMH was made possible by the discovery of certain hydrogen storage alloys that could store hydrogen in the interstitial sites of the alloys without forming overly strong metal–hydrogen bonds [1]. Thus, a solid-state hydrogen electrode could be created. The Ni electrode was the same as in the NiCd battery. The NiMH battery chemistry is, however, simpler than the NiCd chemistry in that hydrogen is directly available at the metal hydride alloy surface. In the NiCd battery hydrogen has to be created by corroding the cadmium metal in the alkaline electrolyte. This facilitates the construction of the NiMH cell as the volume of the electrolyte is constant over the charge/discharge cycle in the NiMH cell in contrast to the NiCd cell [1]. The Li cell has a few things in common with the NiMH cell. Both have electrodes based on intercalation reactions and no dissolution/precipitation reactions, as is the case for lead–acid and NiCd batteries. This opens up for easier control of parasitic reactions and, thus, for a possible long cycle life for both new battery chemistries. In the NiMH cell, hydrogen is shuttled between the electrodes. In the Li cell, lithium is doing the same. In the charged state, hydrogen is stored as a metallic hydride in the NiMH cell, while in the Li cell, lithium is usually stored in graphite or coke. When the cells are discharged, hydrogen and lithium are transferred to the counter electrodes consisting of transition metal oxides. In the NiMH cell, they were nickel oxide, and for the first commercial Li cells, they were cobalt oxide. In oxygen-containing electrodes, the hydrogen and lithium are firmly bonded as H^+^ and Li^+^ ions. It is essentially these bonds that contain the stored energy in the cells. The stronger bond in the Li battery compared to the oxygen–hydrogen bond in the NiMH battery gives the Li battery the benefit of having a three times higher voltage, approximately 3.7 V compared to 1.25 V for the NiMH cell. However, the Li cell capacity is not three times higher as the metal hydride electrode can store a high density of hydrogen atoms. The lower voltage in the NiMH battery brings advantages as it allows the use of a highly conducting water-based alkaline electrolyte, which also simplifies the choice of separator. In the Li cell, non-aqueous electrolytes with significantly poorer conductivity have to be used, forcing the separator to be very thin to compensate for the poor conductivity. The micron-thin separator that separates the highly reactive electrode materials leads to stringent quality demands for safety reasons. The presence of hydrogen in the metal hydride electrode further mitigates corrosion of the hydrogen storage alloy. If hydrogen in the NiMH cell is corroded, it forms water, which is the base for the electrolyte. In the Li cell, on the other hand, the very reactive lithium itself is prone to corroding and forming insolvable species with a negative influence on performance. 

As explained later, we showed that the good corrosion stability of the metal hydride alloy could be used to multiply the cycle life of NiMH cells by the addition of oxygen and hydrogen gas [2]. This will make NiMH batteries more competitive in relation to Li batteries. The high capacity of Li batteries is certainly a big advantage but, in many applications, it is more important to both store and deliver a high amount of energy throughout the battery lifetime. This means that capacity times cycle life will be an important parameter when comparing batteries. Another issue to compare is how easy it is to recycle the used batteries. In the present article, it is suggested that the good corrosion stability of the MH alloy can be turned into a very simple recycling process based on a simple mechanical washing and rinsing process. 

Modern metal hydride (NiMH) battery alloys are produced by a series of processes that have been continuously developed since NiMH batteries were commercialized at the beginning of the 1990s. The metals are melted and mixed in a crucible held under a protective atmosphere, while composition and temperature are carefully controlled. The alloy is then cast in a rapid solidification process. More recently, this has started to be done by strip casting. Finally, an annealing step is applied to promote uniformity before the final alloy is pulverized and sieved. The development of these processes over time has resulted in robust and corrosion-resistant metal hydride storage alloys for NiMH batteries. Despite the development of new and improved materials, the corrosion of the metal hydride is still indirectly detrimental for life expectancy, as the corrosion consumes the aqueous electrolyte. The loss of electrolyte increases the internal resistance, making the cells incapable of working at high currents. This is aggravated by the fact that the cells are filled with a very limited amount of electrolyte in order to allow oxygen and hydrogen gas to pass between the electrodes through openings in the porous separator. This facilitates overcharge and over-discharge protection reactions via the gas phase. The arrangement is usually called “a starved electrolyte concept” [1,2].

When the cell is charged, Ni(II)(OH)_2_ is converted to Ni(III)OOH and hydrogen is transferred from the Ni electrode to the MH electrode in the form of water molecules after reacting with OH^−^ ions. When the 2-valent Ni(OH)_2_ is depleted of available hydrogen, the cell goes into an overcharge mode, and hydrogen is taken from the electrolyte, leading to oxygen gas evolution at the Ni electrode. If the separator between the electrodes is not completely filled with electrolyte, the oxygen molecules can pass through open channels in the separator and recombine with hydrogen to form water at the metal hydride electrode. This leads to a steady-state charge level and a temperature rise, which can be detected and used for terminating the charge. When the cell is discharged, OH^−^ ions react with hydrogen from the metal hydride forming a water molecule. Concomitant at the Ni electrode a water molecule will be deprived from a hydrogen atom reforming Ni(II)(OH)_2_ and an OH^−^ ion. When the cell is over-discharged and the Ni(OH)_2_ cannot accept more hydrogen, a similar process will recombine gaseous hydrogen into water. Both these recombination reactions mitigate the pressure increase in the cells and prevent the safety valve from opening, which can lead to an accelerated loss of electrolyte. The corrosion of the metal hydride alloys by the alkaline electrolyte, however, also evolves H_2_ gas when the metals form hydroxides in the corrosion process. When this hydrogen is absorbed in the MH electrode, it shifts the electrode balance in the cells, exhausting the overcharge reserve. This causes the overcharge reactions to stop working and if the pressure increases and vents through the safety valve, the cells quickly dry out [3].

Initial corrosion is, however, necessary for creating an active surface that can promote the kinetics of the electrode reactions. Most NiMH batteries are based on LaNi_5_ or La(Mg)Ni_7_ derivatives, commonly labeled AB_5_- or A_2_B_7_-type alloys [4]. A is a mixture of electropositive metals, usually dominated by rare-earth (RE) metal mixtures corresponding to the composition of the source from where it is mined. B is essentially Ni partly substituted by Co, Mn, and Al to adjust the hydrogen uptake and release properties. The corrosion of the alloy yields needle-like crystals of RE hydroxide adhered to the outermost surface. A denser underlying surface coating acts as an interphase between the electrolyte protecting the inner metallic part of the alloy particle. This protective interphase also holds nickel-containing nano-sized clusters formed during the initial corrosion. The clusters are important for the catalytic activity promoting the making and breaking of O–H bonds during discharge and charge, respectively [5,6].

It is, however, also important that the protective coating is stable against further corrosion, so it can continue to protect the inner metallic part that stores hydrogen [2]. In a recent paper, it was confirmed that the corrosion protection by the initial corrosion during cycling is actually rather effective [2]. The drying out of the cells was mitigated by adding pure oxygen gas to an aged NiMH battery pack, to see if the oxygen could react with the hydrogen liberated in the preceding corrosion process. This worked well, as the oxygen reacted with hydrogen without corroding the metals, forming water that could replenish the electrolyte. The oxygen addition also reestablishes the electrode balance. The performance of the used battery recovered with respect to internal resistance and capacity because of the replenished electrolyte. This indicated that most of the alloy is well-protected by the surface in a spent NiMH battery after the battery has reached its end of life [2]. In another recent paper, the presence of rare-earth metals was suggested as a reason for the beneficial corrosion stability [7]. This provided the inspiration for trying a new method for regeneration of MH electrode materials by simply cleaning and washing the powder surfaces. Thus, energy-consuming resmelting and recasting of MH alloys could be avoided, as well as complicated chemical-consuming hydrometallurgical processes used in conventional recycling of battery materials.

In this work, discharged bipolar NiMH 12-volt battery modules produced by Nilar AB in Sweden were disassembled, and the spent MH electrodes after the modules had been in use for long-term testing were collected. Nilar makes MH electrodes by pressing a dry mixture of hydrogen storage alloy and Ni powder on to a polymeric carrier. No substrates such as Ni-foam or expanded metal are used. This facilitates a simple retrieval of the electrode materials.

To recover the performance of the spent alloy, mechanical as well as chemical treatments were tested in an effort to remove the corrosion products. Mechanically, sonification and ball milling combined with a simple washing of the resulting mixture were tested.

Sonication provides a flexible way of treating a system with respect to time of interaction, pressure, and applied energy [8]. With a high-power ultrasonic source these characteristics are even further emphasized, making it an interesting tool. As suggested by Chen et al., it may be possible to work upon the hydroxide layer by sonication of either the complete battery cells or, for selective regeneration, the MH electrode material itself [9].

The chemical treatments were composed of washing with acidic or alkaline solutions. As RE hydroxides can be dissolved by acidic solvents, acid treatment is a possible way to remove the passivating RE hydroxide layer. Sulfuric acid leaching has been used for RE recovery from spent NiMH batteries [10,11,12]. It is, however, a complicated process. Here, a low-concentration sulfuric acid solution was tested for the removal of the corrosion layer from the bulk alloy. The dissolution of the RE hydroxides can be described as:2RE(OH)_3_ (s) + 3H_2_SO_4_ (aq)→ RE_2_(SO_4_)_3_ (aq) + 6H_2_O(1)

Hot alkaline treatments have been found to improve the electrochemical performance of AB2 and AB5-type hydrogen story alloys [13,14]. Therefore, there is the possibility of washing away the hydroxide layer by cooking in a hot alkaline solution.

## 2. Result and Discussion

The rare-earth hydroxide needles formed on the MH particles can be removed by sonication. Figure 1 shows the SEM images of the cycled and sonotrode-sonicated samples. After 2 h of sonication, the alloy showed a very clean surface. Figure 2 shows X-ray diffraction patterns of cycled alloy and sonotrode-sonicated alloys. With increasing sonication time the peaks of rare-earth hydroxide decreased. The peaks were, however, still present even if the Mm(OH)_3_ needles were not observed in the SEM. This indicates that a bulk hydroxide interphase formed underneath the needles remains. From Figure 3, the half-cell test shows an optimum in capacity (~240 mAh g^−1^) for the sample with 30 min of sonication, however, the discrepancy is still large when compared to the fresh alloy, which had a capacity around 320 mAh g^−1^ [5]. This is probably due to a thicker interphase covering the surface as the corrosion-resistance of the alloy is lower. The sample with 2 h of sonication showed a lower capacity, possibly due to a passivation of the active catalytic species in the interphase.

Inspired by this, large-scale sonication in an ultrasonic bath with a lower power input than the sonotrode was tested, with the aim of sparing the catalytic properties. Additionally, a mild ball-milling procedure was tested.

Figure 4 shows the SEM images of cycled, ultrasonicated, and ball-milled material. After the ultrasonic bath, the material had a relatively smooth surface with an overlay of small particle fragments as well as some needle-like structures. The ball-milled material showed a very similar surface to that of the ultra-sonicated sample, but with smaller particle fragments and needle-shaped structures partially covering the surface. This indicates that the rare-earth hydroxide layer can be partly removed in an ultrasonic bath as well as by ball-milling, as shown in Figure 5, where the peaks of rare-earth hydroxide slightly decrease after treatments.

Cross-sections of the untreated cycled material and the ball-milled material were prepared (Figure 6). The ball-milled material showed a mixture of particles with fresh surfaces from particles cracked in the ball-milling and corroded surfaces, where the ball-milling had just separated the particles and removed most of the rare-earth hydroxide needles. This contrasts with the cycled material, where a corrosion layer could be seen on all surfaces throughout both fissures and the exterior. The surface layer observed on the corroded particles was determined to be oxygen-rich by EDS mapping (Figure 7). This oxygen-rich structure is most likely, as previously shown, made up of rare-earth hydroxides [5].

In Figure 8 we can see that the half-cell measurement of the cycled alloy straight out of the spent battery lost about 5% of the initial capacity, showing that most of the alloy capacity remains even if the battery has passed the end-of-life criteria caused by electrolyte dry-out. The balled-milled and washed alloys showed the best performance. Not only did they reach the capacity of the pristine alloy but they also showed a very rapid activation. This indicates that the active surface created by the corrosion during the formation of the original battery alloy was still in most aspects intact. The ultrasonic bath-sonicated sample showed a similar behavior even if it did not reach the same capacity, probably due to a less effective removal of the rare-earth hydroxide corrosion products.

The tendency towards improved and quicker activation was also reflected in the high rate discharge properties (Figure 9). As a result of the cycling during the previous long-term testing, the alloy particles formed not only an active surface but they also cracked up into smaller particles, thus increasing the surface area. As such, the reaction kinetics were not only improved by the presence of a reactive surface, but also by a more accessible surface. As seen in Figure 6, however, a number of those particles did not separate properly, leading to long diffusion paths in the electrolyte. Sonication and ball-milling help to open up the agglomerated particles, resulting in better accessibility for the ionic transport.

The fragmentation of the particles by ball-milling and sonication also changed the particle size distributions (Figure 10), where the smaller particle sizes in the ball-milled and bath-sonicated materials corresponded to a larger active surface.

The same characterizations were performed on acid-treated samples. Figure 11 shows SEM micrographs of the cycled alloy before the acid treatment and of samples treated with different concentrations of sulfuric acid. The layer of needle-shaped crystals was completely removed when using H_2_SO_4_ solutions with concentrations of 0.05 M to 1 M (Figure 11b–d). However, with high acid concentration, the bulk alloy was attacked (Figure 11d). The result was confirmed by XRD (Figure 12). Compared to the original cycled alloy, the rare-earth hydroxide peaks of samples after acid treatments were non-existent. The discharge curves from half-cell tests (Figure 13) showed an optimum capacity for the sample treated with the lower concentration of H_2_SO_4_. The sample which was treated in 0.05 M H_2_SO_4_ for 1 min also showed a very rapid activation (Figure 14). This indicates that the active surface created by the corrosion during the formation of the original battery alloy was still in most aspects intact (as we described for ball-milled sample above). Samples treated with higher concentration of H_2_SO_4_ (0.1 M and 1 M) showed slow activation, because the active surface/interphase was probably damaged. This indicates that the presence of an interphase was important for good performance, supplying both corrosion protection as well as catalytically active species. The results indicate that the RE hydroxide is very quickly dissolved with dilute sulfuric acid solutions, but also that it is difficult to design practical procedures to obtain optimal performance with a suitable thickness of the passive surface layer.

Hot KOH was also evaluated as a possible washing media. Three different washing times were tested (5 min, 10 min, and 20 min) but the hydroxide layer could not be removed. The layer grew even thicker in hot KOH, as can be seen from the SEM images (Figure 15) and XRD pattern (Figure 16). During the treatment, the alloy cracked into smaller particles and hydroxide layers formed on the new surfaces. Also, the size of the needle-shaped hydroxides increased in the KOH. Figure 16 shows the discharge curves of alkaline-treated samples. There was not much change in capacity with 20 min of hot KOH boiling. In addition, the resistance increased, as seen by the lower voltage in Figure 17, indicating a poorer contact resistance caused by the additional corrosion. Therefore, hot alkaline washing does not seem to be a good method to regenerate spent alloy.

## 3. Materials and Methods

### 3.1. Mechanical Treatments

#### 3.1.1. Sonication

The sonication in this work was performed by using either a sonotrode or an ultrasonic bath. MH electrode material (2 g) was sonicated for 2 h using a 14 mm sonotrode (Hielscher UP200Ht) immersed in 70 mL of deionized (DI) water in a 100 mL beaker. Thereafter, the sample was washed with a total of 1 L of DI water and vacuum-dried for 2 h. Each sonication had a setting of 100% amplitude, giving a total power input of 200–4200 mWh mL^−1^. An AB_5_-type alloy, where the alloy producer indicated the composition to be MmNi_4.08_Mn_0.46_Co_0.46_Al_0.32_, constituted this MH electrode material, where “Mm” stands for a lanthanum–cerium rich rare-earth mixture.

In the ultrasonic bath, 5 g of cycled anodic material was put in a 25 mL glass beaker containing 15 mL of deionized (DI) water. The glass beaker was then suspended halfway into the ultrasonic bath and treated for 30 min, giving a power input of approximately 160 mWh mL^−1^. After the treatment, the acquired light-gray suspension was poured off and the remaining powder was rinsed with 10 mL of DI water and dried under vacuum for 4 h. The mass loss during the ultrasonic treatment was approximately 0.9 wt.% (dry mass before and after treatment). Here a more cobalt-containing and thus more corrosion resistive alloy was used. The composition of this lanthanum–cerium-based AB_5_ alloy is MmNi_3.61_Mn_0.33_Co_0.68_Al_0.29_, as specified by the supplier. Both these alloys were also investigated in [2].

#### 3.1.2. Ball Milling

Approximately 20 g of material was high-energy ball-milled for 15 min with 10 mL of DI water, giving a dark-gray suspension above a layer of electrode material. The mixture was then transferred to a 250 mL beaker and stirred for 1 min. It was then left to settle for one minute to achieve a rough separation, with larger and/or denser particles precipitating onto the bottom of the beaker. The liquid content and the still suspended particles were poured off (saved for further analysis) and the remaining thick slurry was transferred to a glass frit and washed with 50 mL of DI water. It was then dried under vacuum for 4 h, yielding a fine metallic powder. The material loss corresponded to 1.8 wt% of the initial weight (dry mass before and after treatment). The same alloy as in the ultrasonicated bath was used (i.e., with composition MmNi_3.61_Mn_0.33_Co_0.68_Al_0.29_).

### 3.2. Chemical Treatments

#### 3.2.1. Acid Treatment

Acid treatment was applied using 0.05 M, 0.1 M, and 1 M sulfuric acid solutions. The treatment time was 1 min. The solutions were prepared from 95% H_2_SO_4_ (VWR Chemicals, AnalaR NORMAPUR). The treated material (2 g) was subsequently washed in steps with 1.5 L of DI water (giving a pH = 7) and dried under vacuum for 2 h. The alloy with the composition MmNi_4.08_Mn_0.46_Co_0.46_Al_0.32_ was used in both the acid treatment and the alkaline treatment.

#### 3.2.2. Alkaline Washing

The spent alloy (2 g) was heated in a hot 6 M potassium hydroxide solution at 90 °C for 5 min, 10 min, and 20 min, respectively, after which the powders were washed with deionized water and dried in vacuum for 2 h.

### 3.3. Morphological Analysis and Structural Characterization

Powder X-ray powder diffraction patterns for all samples were collected with a Panalytical X’pert Pro diffractometer (Malvern Panalytical Ltd., Malvern, United Kingdom) using CuKα radiation, (λ = 1.5418740 Å). Scanning electron microscopy (SEM) micrographs and energy-dispersive spectroscopy (EDS) data were acquired with a JEOL 7401F Field-Emission Scanning Electron Microscope (JEOL Ltd, Tokyo, Japan) and a JEOL 7400F analytical SEM setup (JEOL Ltd, Tokyo, Japan), and a Hitachi TM3000 table top SEM (Hitachi Ltd, Tokyo, Japan) was used for particle size assessments. For the cross-section imaging, the powder samples were mixed with a conductive carbon cement and polished using a JEOL SM-09010-CP cross-section polisher.

### 3.4. Electric Characterization

The discharge capacities of the regenerated alloy were measured in half-cell tests performed with a Lanhe CT2001A (Wuhan LAND Electronic Co. Ltd, Wuhan, China) battery testing instruments using a 6 M KOH electrolyte. At least 3 half-cells were made from each sample. Regenerated alloy (0.25g) was mixed with 0.75 g of copper powder (200 mesh) and then compacted in a pressing tool with a 1 cm diameter to a pellet. The pellet was wrapped in a nickel net (100 mesh) and pressed again. A nickel wire spot-welded to the net was used as current collector. This method of making working electrodes was described in a previous paper [5]. We used a 20-mesh nickel net spot-welded to a nickel wire as counter electrode, and a 3 mm Zn-rod as a reference electrode. The working electrodes were charged at 15 mA for 6 h (corresponding to a maximum capacity of 360 mAh g^−1^) and then discharged at 0.2 C for 10 cycles. Samples treated by ultrasonic bath and ball-milling were discharged at 1 C, 2 C, 4 C, and 8 C for 5 cycles per discharge rate (1 C = a current equal to the nominal cell capacity divided by 1 h, i.e., a current that will discharge the cell in 1 h). The cut-off voltages were +0.75 V, +0.9 V, +1.1 V, and +1.2 V vs. Zn/Zn^2+^, respectively.

## 4. Conclusions

Mechanical and chemical treatments as methods to recover hydrogen storage alloys from spent NiMH batteries were evaluated.

The corroded surfaces of the cycled alloy particles were confirmed by XRD to contain rare-earth hydroxides which could be partially removed by mechanical treatments while maintaining the catalytic properties of the alloy particles. The good reaction kinetics were attributed to nickel-containing clusters in the interphase covering the inner metallic part of the alloy powder. This indicates that the active clusters in the surface were not destroyed. Sonication and ball-milling also helped to break-up agglomerated particles, thereby improving the high-rate properties of the alloy by increasing the access to active surface sites. A subsequent washing step separated the active powder from the corrosion products. This was facilitated by utilizing their different sedimentation properties.A suitable acid leaching method could also remove the corroded layer. However, this was difficult to control and the whole interphase could accidentally be lost.Treatment with a hot alkaline solution made the hydroxide layer grow thicker and did not improve properties.Reuse of ball-milled or sonicated material could serve as a simple recycling alternative to energy-demanding metallurgical smelting methods and chemical-consuming hydrometallurgical recycling processes, where the ease of use and possibilities of up-scaling further favor the less complex mechanical treatments.The acquired material, upon partial or full reimplementation into NiMH battery production, could provide a longer cycle life as it has already been activated and partially corroded, which means that the alloy will consume less of the electrolyte during the initial formation of the cells. Improved high-rate properties of the anodic material could also be expected. It is also expected that the cycling stability would be improved with an already-formed surface in the upcycled alloy.

The removal of corrosion products and smaller particle fragments provides an increased capacity. Since the spent material already has active surfaces, the need for activation of the material is reduced, as shown in the half-cell tests. The stable but catalytic interphase protecting the inner particles indicates that the MH electrode material may perform better in its second life in a new NiMH battery. It is also expected that the cycling stability will be improved due to an already formed surface in the upcycled alloy. This will be tested in the future. Further trials should consist of performing more precise density separation as well as scaling up the amount of material treated. The material should also be tried in full cell battery production to test the hypothesis of an increased cycle life as well as the possibility of faster activation and improved high-rate properties.

## Figures and Tables

**Figure 1 molecules-25-02338-f001:**
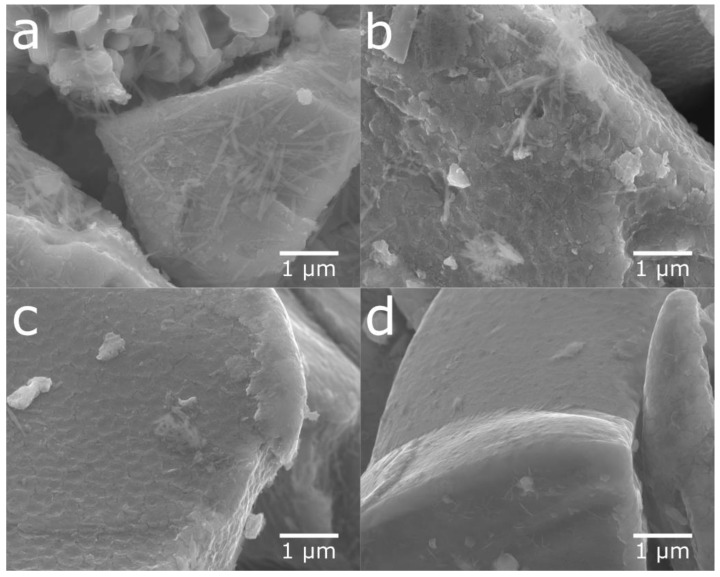
SEM secondary electron images at 20,000 times magnification. Sonotrode-sonicated samples of cycled material at (**a**) t = 0, (**b**) t = 5 min, (**c**) t = 30 min, and (**d**) t = 2 h. The amount of needle-like rare-earth crystals laying parallel to the metal surface in (**a**) is seen to decrease when going from (**a**) to (**d**).

**Figure 2 molecules-25-02338-f002:**
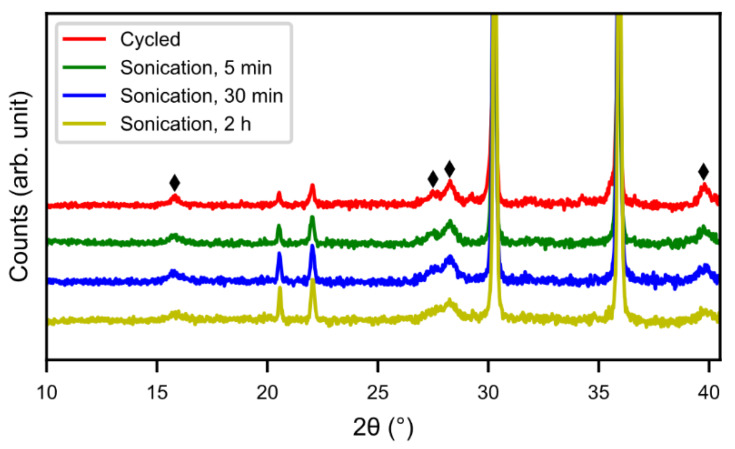
XRD patterns of the sonotrode-sonicated cycled alloy at 5 min, 30 min, and 2 h, respectively. (♦) RE(OH)_3_.

**Figure 3 molecules-25-02338-f003:**
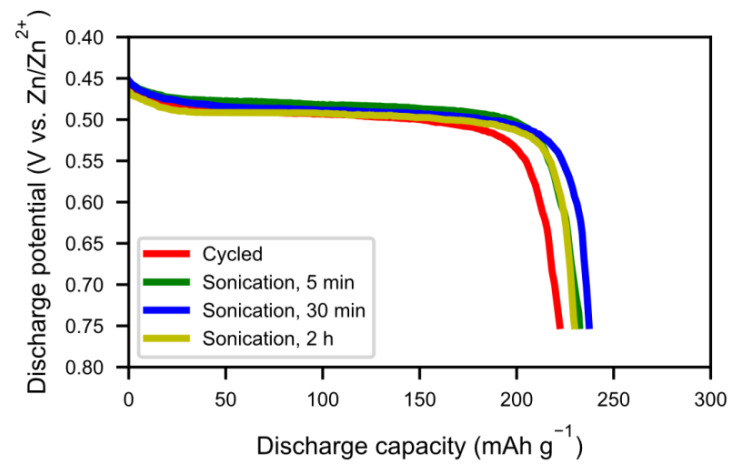
The discharge curves at 60 mA g^−1^ for the alloy with 5 min, 30 min, and 2 h of sonotrode-sonication, as described in Figure 1 and Figure 2.

**Figure 4 molecules-25-02338-f004:**
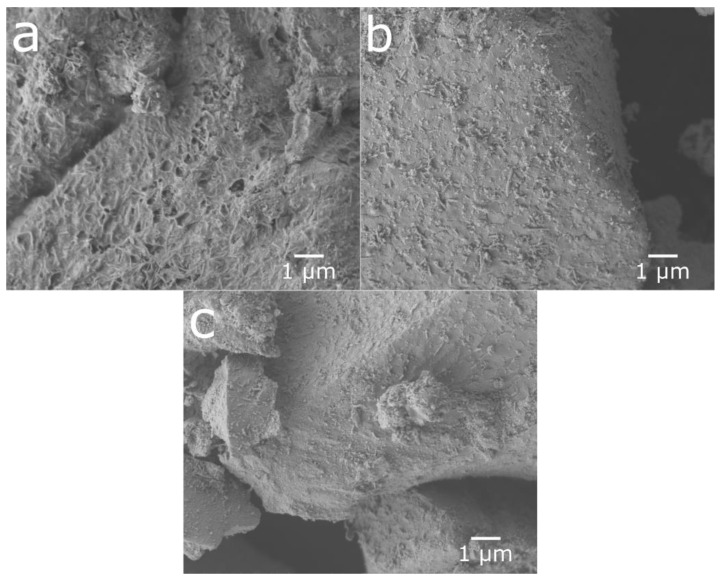
SEM micrographs of particle surfaces from (**a**) cycled material, (**b**) material from the ultrasonic bath, and (**c**) ball-milled material.

**Figure 5 molecules-25-02338-f005:**
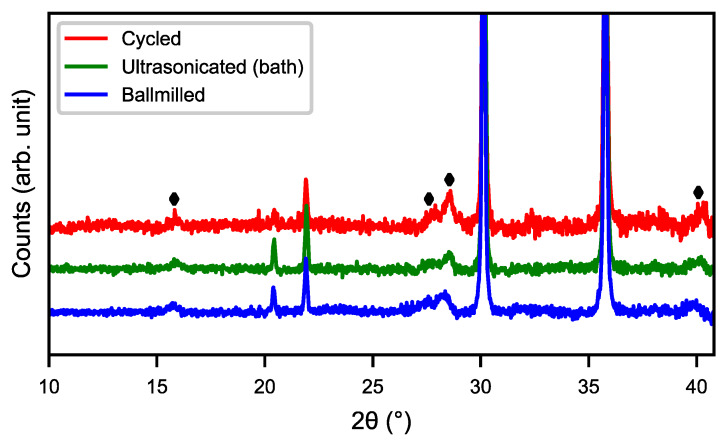
XRD patterns of cycled material, material from the ultrasonic bath, and ball-milled material, respectively. (♦) RE(OH)_3_.

**Figure 6 molecules-25-02338-f006:**
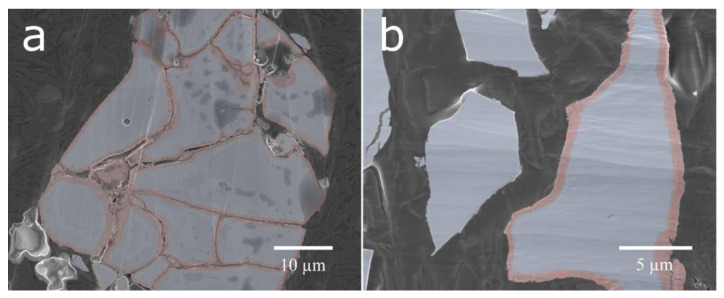
Cross-sectional micrographs of (**a**) cycled material and (**b**) ball-milled material.

**Figure 7 molecules-25-02338-f007:**
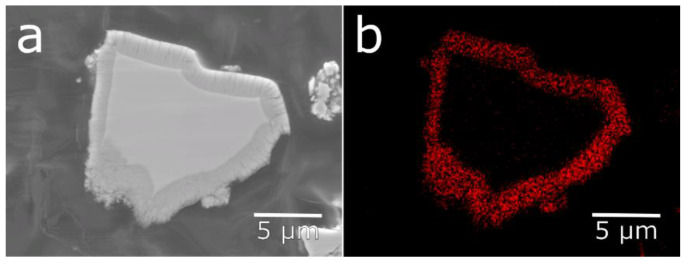
SEM image (**a**) and EDS map with oxygen Kα counts (**b**) of a corroded alloy particle, showing a surface layer with elevated oxygen content.

**Figure 8 molecules-25-02338-f008:**
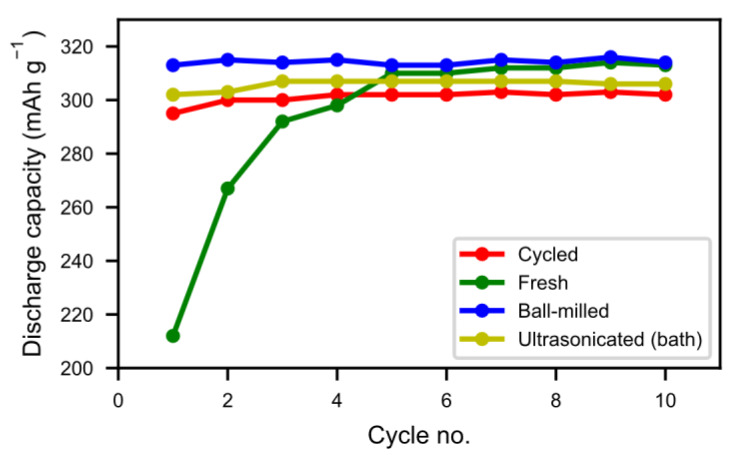
Average capacity measured for the first 10 cycles of the treated and untreated samples. For all 10 cycles, a discharge rate of 60 mA g^−1^ (0.2 C) was used. The measured capacity was adjusted to take the added Ni in the electrode fabrication into account.

**Figure 9 molecules-25-02338-f009:**
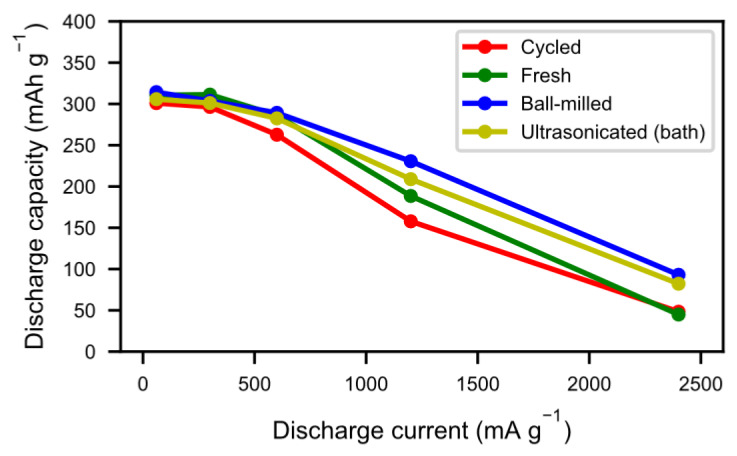
The discharge capacity at 60 mA g^−1^, 300 mA g^−1^, 600 mA g^−1^, 1200 mA g^−1^, and 2400 mA g^−1^ for fresh material, cycled material, ball-milled material, and bath-ultrasonicated material.

**Figure 10 molecules-25-02338-f010:**
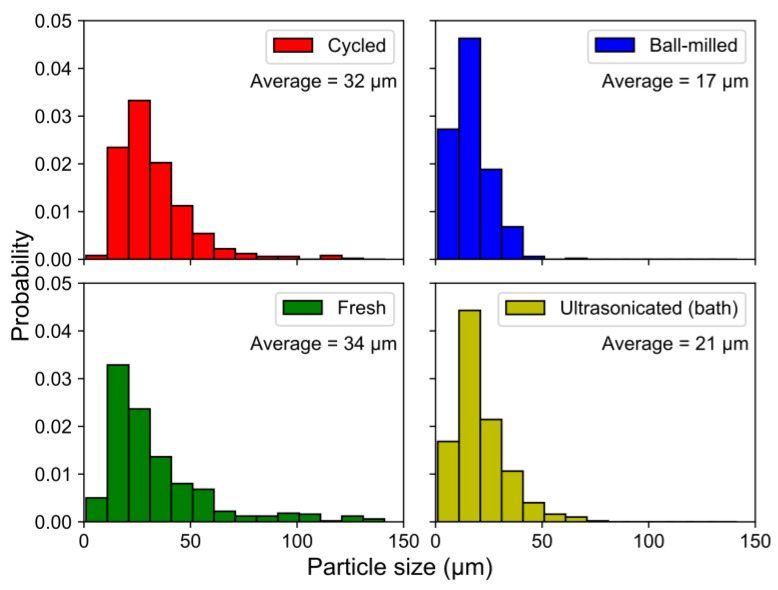
Particle size distributions for the treated/untreated samples, measured by SEM. For each sample, a total of 500 measurements were taken. A significant reduction in the average particle size can be seen for the treated materials.

**Figure 11 molecules-25-02338-f011:**
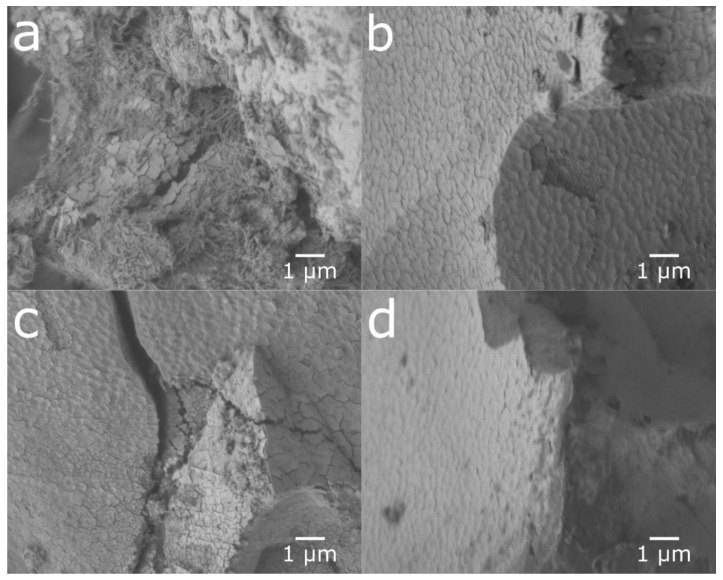
SEM secondary electron images at 10,000 times magnification. Acid-treated samples of cycled anodic material: (**a**) untreated, (**b**) 0.05 M H_2_SO_4_ for 1 min, (**c**) 0.1 M H_2_SO_4_ for 1 min, (**d**) 1 M H_2_SO_4_ for 1 min.

**Figure 12 molecules-25-02338-f012:**
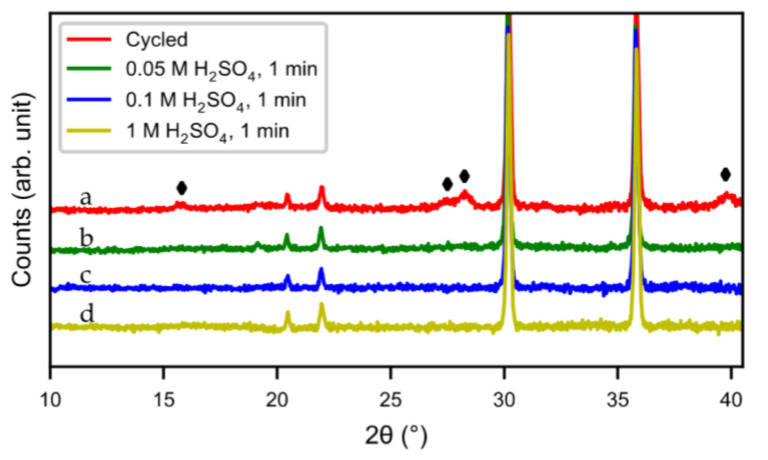
XRD patterns of acid-treated samples, (**a**) untreated, (**b**) 0.05 M H_2_SO_4_ for 1 min, (**c**) 0.1 M H_2_SO_4_ for 1 min, (**d**) 1 M H_2_SO_4_ for 1 min. (♦) RE(OH)_3_.

**Figure 13 molecules-25-02338-f013:**
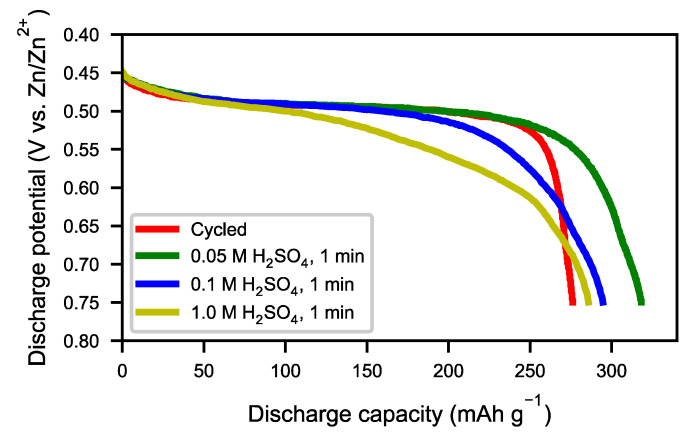
The discharge curves at 60 mA g^−1^ of cycled anodic material: cycled alloys treated with 0.05 M H_2_SO_4_ for 1 min, 0.1 M H_2_SO_4_ for 1 min, and 1 M H_2_SO_4_ for 1 min, respectively.

**Figure 14 molecules-25-02338-f014:**
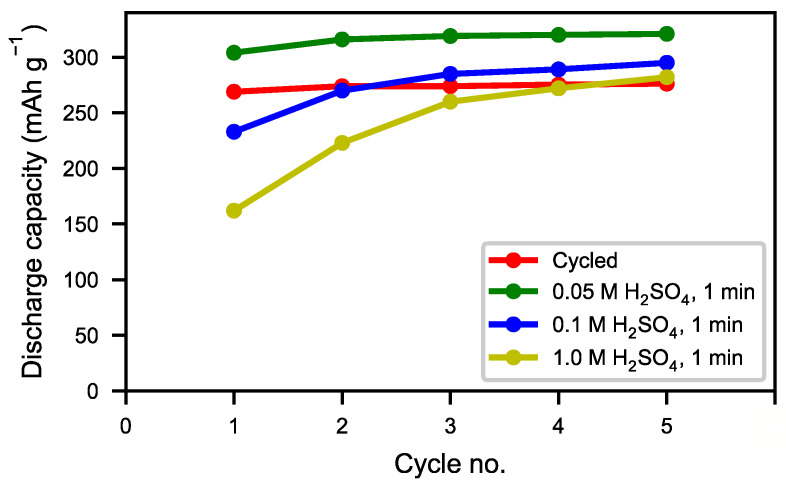
Discharge capacity measured for the first 5 cycles for the treated and untreated samples. For all 5 cycles, a discharge rate of 60 mA g^−1^ (0.2 C) was used. The measured capacity was adjusted to take the added Ni in the electrode fabrication into account.

**Figure 15 molecules-25-02338-f015:**
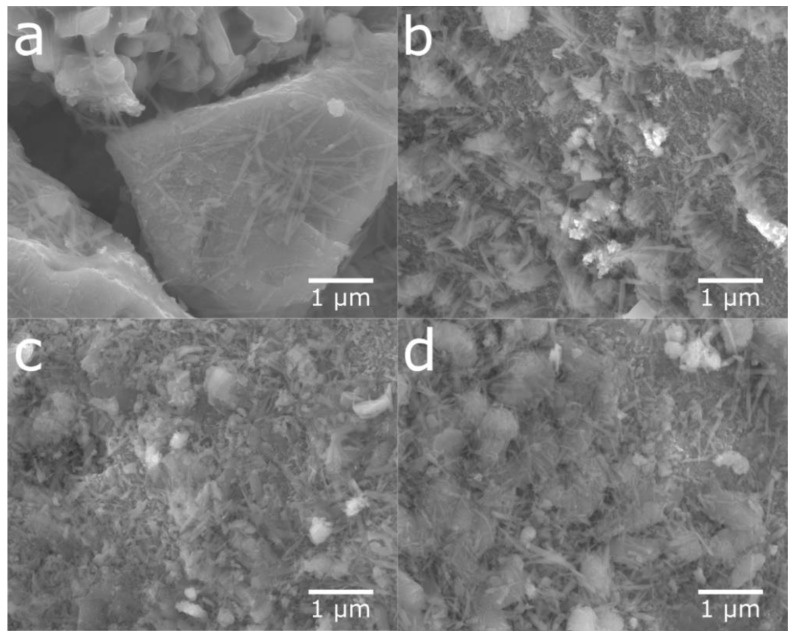
SEM secondary electron images at 20,000 times magnification. Hot KOH treatment of cycled metal hydride (MH) material: (**a**) t = 0, (**b**) t = 3 min, (**c**) t = 10 min, and (**d**) t = 20 min.

**Figure 16 molecules-25-02338-f016:**
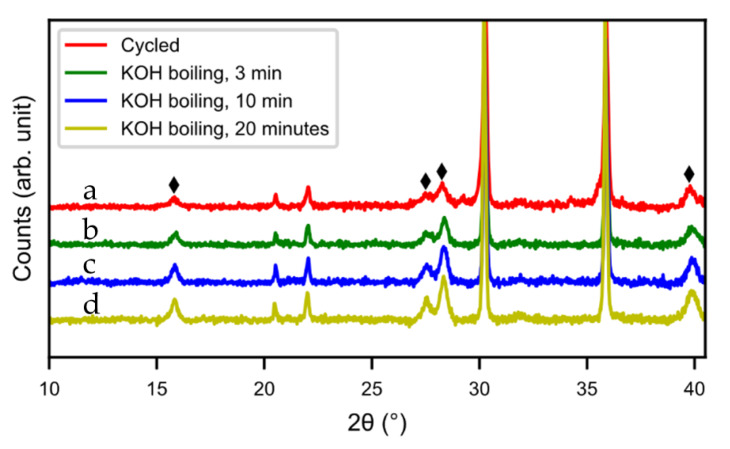
XRD patterns of alkaline-treated samples, (**a**) untreated, (**b**) t = 3 min, (**c**) t = 10 min, (**d**) t = 20 min. (♦) RE(OH)_3_.

**Figure 17 molecules-25-02338-f017:**
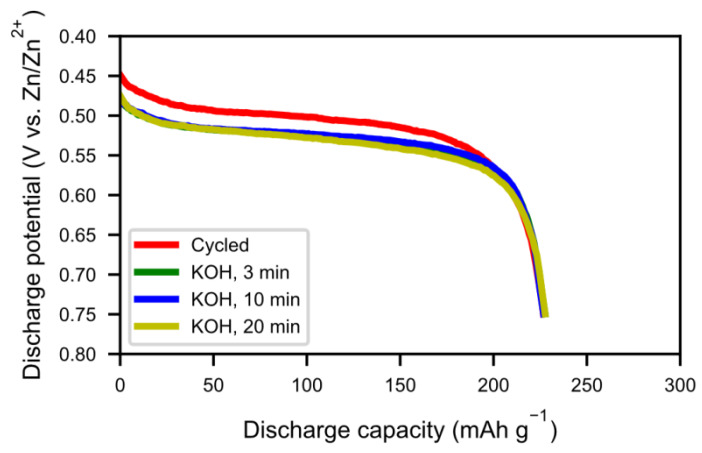
The discharge curves at 60 mA g^−1^ for spent alkaline-treated cycled alloy at 3 min, 10 min, and 20 min, respectively.

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
