# Peer review of "Upcycling of Spent NiMH Battery Material—Reconditioned Battery Alloys Show Faster Activation and Reaction Kinetics than Pristine Alloys"

_molecules, 2020, doi:10.3390/molecules25102338_

Round 1
Reviewer 1 Report
The topic of the paper, i.e. the possibility to recycle hydrogen storage alloys from spent NiMH batteries and the study of the behavior of the MH-electrode material after the recycling mechanical treatments, is very interesting and fashionable. The techniques used for "re-conditioning" the material and to characterize it are very suitable and useful and the results are well commented and critically revised. The conclusions are well supported by the results. The figures are clear and well edited.
My suggestions are about:
- the English language. I suggest to read again the paper since some English mistakes are present, i.e. singular forms in place of plural forms and vice versa (example: Both these recombination reactions mitigates the pressure increase); and sentences written like a laboratory book and not suitable for a scientific publication (example Acid treatment applied, 0.05M, 0.1M and 1M sulfuric acid solutions).
- the use of the active form of the verbs and the "we form". In general, passive forms of the verbs are used in a scientific publication (it was found, it was described in place of We found). I suggest to change these verbs and these expressions.
- the paragraph "Analysis Morphological and Structural Characterization". I suggest to change the title in "Morphological analysis..." and to better specify the conditions and the parameters used for the presented measurements (i.e. kind of crucibles and sample holders; measurement atmosphere - air or inert gas -, scanning time and angle for XRPD, working distance for SEM).
- The literature references. Some of them appear in the text as "Error! Bookmark not defined". They must to be checked and re-inserted.
Author Response
Comments to the reviewers in red in the appended file:

Reviewer 2 Report
This is an interesting article about recycling of NiMH battery alloy material. The Ni hydride has been recycled through various means - sonication, ball milling, acid treatment and alkaline washing. After each treatment the surface has been studied by SEM and the phase composition of the surface determined by XRD. After each treatment the discharge capacities of the material was evaluated.
The manuscript needs following revisions:
(1) reference numbers: the numbers should be placed before the full stop of the sentence, not after. And, I suppose this is the style of the journal, multiple references be put in one set of brackets "[10,11]." On page 3 line 128, the reference should be placed in brackets "... in reference [2]."
Some of the reference numbers do not appear [Error...]
(2) In the experimental section, since all the parts are placed under a heading, measurement of discharge capacities should also be given a heading.
(3) Some sentences are not full, such as page 4 line 150 "Analysis of Morphological..." In this section there are some more such sentences.
(4) The needle-like morphology on surface in Fig. 1(a) should be pointed out to the readers. In the caption of this figure, "20000 times magnification" does not mean enough because we do not know at what magnification this figure is seen. The micron scale (1 um) in micrographs should be clearly visible to the readers.
(5) On page 8, "fresh and corroded surfaces" should be clearly defined, even if redundantly, to avoid any confusions during reading. Similarly, "untreated cycled alloy" on page 9, and any other in the text.
(6) Fig. 13 a and b put together with one caption.
(7) The conclusions reads like a long summary. Main conclusions should be put clearly point by point to the readers .
(8) Some others such as on page 1 line 34: "The metals are melted"
Author Response
Comments to reviewer in red
